# RNA-Based Strategies for Cancer Therapy: In Silico Design and Evaluation of ASOs for Targeted Exon Skipping

**DOI:** 10.3390/ijms241914862

**Published:** 2023-10-03

**Authors:** Chiara Pacelli, Alice Rossi, Michele Milella, Teresa Colombo, Loredana Le Pera

**Affiliations:** 1Department of Biochemical Sciences “A. Rossi Fanelli”, Sapienza University of Rome, 00185 Rome, Italy; 2Section of Oncology, Department of Medicine, University of Verona-School of Medicine and Verona University Hospital Trust, 37134 Verona, Italy; 3Institute of Molecular Biology and Pathology (IBPM), National Research Council (CNR), 00185 Rome, Italy; 4Core Facilities, Italian National Institute of Health (ISS), 00161 Rome, Italy

**Keywords:** targeted exon skipping, antisense oligonucleotide, ASO, splicing, RNA therapy, cancer, bioinformatics

## Abstract

Precision medicine in oncology has made significant progress in recent years by approving drugs that target specific genetic mutations. However, many cancer driver genes remain challenging to pharmacologically target (“undruggable”). To tackle this issue, RNA-based methods like antisense oligonucleotides (ASOs) that induce targeted exon skipping (ES) could provide a promising alternative. In this work, a comprehensive computational procedure is presented, focused on the development of ES-based cancer treatments. The procedure aims to produce specific protein variants, including inactive oncogenes and partially restored tumor suppressors. This novel computational procedure encompasses target-exon selection, in silico prediction of ES products, and identification of the best candidate ASOs for further experimental validation. The method was effectively employed on extensively mutated cancer genes, prioritized according to their suitability for ES-based interventions. Notable genes, such as NRAS and VHL, exhibited potential for this therapeutic approach, as specific target exons were identified and optimal ASO sequences were devised to induce their skipping. To the best of our knowledge, this is the first computational procedure that encompasses all necessary steps for designing ASO sequences tailored for targeted ES, contributing with a versatile and innovative approach to addressing the challenges posed by undruggable cancer driver genes and beyond.

## 1. Introduction

Cancer, the second leading cause of mortality worldwide, is expected to increase by more than 60% in the next two decades, posing a significant public health challenge [1]. Over the past few decades, significant advancements in DNA sequencing and a more comprehensive understanding of the cancer genome have revolutionized the cancer treatment landscape. The traditional paradigm, according to which chemotherapy drugs (chemical substances predominantly targeting rapidly proliferating cells) are selected based on the organ of origin, histology, and staging, has shifted towards the molecular profiling of the tumor (mostly based on genomics) to guide the choice of therapeutic strategies, including chemotherapy, immunotherapy, and molecularly targeted agents [2]. This paradigm shift has paved the way for the emergence of personalized or precision medicine approaches, which hold great promise as effective strategies for treating cancer. Identifying and targeting cancer driver genes [3] containing driver mutations that confer a selective growth advantage to cells is crucial to developing innovative and precision therapeutic approaches. Specifically, these genes can be classified as oncogenes or tumor suppressors based on the type and location of the genetic alterations they undergo. Oncogenes exhibit “gain-of-function” mutations, stimulating cell growth and division, while loss-of-function mutations in tumor suppressor genes lead to the disruption of key checkpoints, such as those regulating cell proliferation, DNA repair, and cell cycle. Taking this into account, classification processes can incorporate various criteria, such as functional studies, somatic mutations, and copy-number alterations [4,5,6]. In particular, the widely used “20/20 rule” combines gain-of-function and loss-of-function mutation occurrence to classify genes as oncogenes or tumor suppressors, as described in the article by Pavel et al., 2016 [7] (see Materials and Methods section for details). Overall, according to the comprehensive Catalogue of Somatic Mutations in Cancer (COSMIC) [8], which provides extensive information on genes with a causal impact on human cancer, there are currently 719 cancer driver genes documented (COSMIC v. 86, August 2018) [9]. This catalog includes details about gene contribution to disease causation, the specific types of mutations that lead to gene dysfunction in cancer, and types of cancer in which mutations have been observed at an increased frequency [10]. Despite the large number of cancer driver genes identified, approved treatments are only available for approximately 40 of these genes [11,12]. Certain genes, such as the RAS family of proteins (KRAS, NRAS, and HRAS, the most frequently mutated oncogenes in cancer [13]), the MYC proto-oncogene (MYC, a commonly amplified gene [14,15]), and tumor protein 53 (TP53, the most frequently altered tumor suppressor gene in human cancer [16]), present significant challenges to pharmacological targeting and have been categorized as “undruggable” [12]. To successfully address these limitations, innovation and technological advancement are necessary [17].

Despite the availability of diverse, promising nucleic acid-based modalities for therapeutic intervention, this work is focused on exploring the potential of antisense oligonucleotide (ASO)-mediated approaches to cancer treatment. This focus is motivated by the greater versatility [18] of ASOs compared, for instance, with small interfering RNAs (siRNAs), which have been also extensively studied in the literature [19,20]. In fact, ASOs not only enable the inactivation of target genes but also offer the ability to modulate protein activity. ASOs could represent a promising class of drugs for personalized medicine approaches, particularly for targeting driver genes that were previously considered unresponsive to traditional therapeutics [21]. ASOs are single-stranded analogs of nucleic acids able to modulate gene expression by selectively binding to target regions through base pairing. Several ASO chemistries have been developed, including 2${}^{\prime }$-O-methyl phosphorothioate (2${}^{\prime }$-OMePS), 2${}^{\prime }$-O-methoxyethyl phosphorothioate (2${}^{\prime }$-MOE-PS), and phosphorodiamidate morpholino oligomer (PMO, also known as morpholinos). These chemistries possess distinct properties aimed at improving the stability, solubility, and cellular uptake of ASOs [22]. They have the ability to interfere with RNA splicing, a crucial step in gene-expression regulation that removes intronic sequences and joins exonic regions to produce mature RNA molecules. Specifically, RNA splicing is catalyzed by the spliceosome, a multimegadalton ribonucleoprotein complex composed of multiple small nuclear RNAs (snRNAs) and many associated protein factors [23]. The spliceosome is recruited through consensus sequence elements at the 5${}^{\prime }$ and 3${}^{\prime }$ splice sites (*donor* and *acceptor* sites, respectively) and branch-point sequences, and its action is further modulated by an array of cis-acting exonic and intronic splicing enhancers (ESEs and ISEs), and exonic and intronic splicing silencers (ESSs and ISSs) [24]. It is precisely by targeting these specific splice sites or splicing regulatory elements that ASOs can induce exon skipping, thereby modulating the splicing outcome [25]. The impact of ASOs on RNA splicing has specific implications in the case of mature transcripts that undergo translation into proteins. An exon-skipping (ES) event can lead to either an in-frame or an out-of-frame transcript, with a potentially significant impact on the protein product and its biological activity [26]. In addition, in the case of an out-of-frame transcript, the loss of the original reading frame may lead to the generation of a premature termination codon (PTC). If the PTC is located at a position sensitive to Nonsense-Mediated Decay (NMD), the transcript may be degraded through NMD, resulting in a loss of protein expression. On the other hand, an in-frame transcript does not affect the reading frame but generates a shorter protein (compared with the original one), which might still retain part of its function. Therefore, the effect of altered skipping on protein synthesis and function depends on the location and size of the skipped exon, as well as on the specific protein and its biological role.

In the last decade, ASOs have shown promising results in the treatment of certain genetic diseases, such as Duchenne muscular dystrophy (DMD; MIM #310200) [27,28]. The FDA has authorized the use of four drugs based on this antisense approach to induce ES for therapeutic purposes in DMD [29,30,31,32]. DMD is a debilitating and progressive neuromuscular disorder that results from mutations in a single gene, the dystrophin gene [33], encoded by a vast locus spanning over 2 million bases on chromosome X and encompassing 79 exons [34,35]. Among the extensive number of annotated mutations (>7000) observed in DMD patients, the majority (∼80%) are large mutations, including deletions of one or more exons (68%) and large duplications (12%), while the remaining ones (∼20%) concern small mutations, such as small in-dels and point mutations [34]. Over 90% of these mutations, which tend to cluster in the region spanning exons 45–53 [36], cause the disruption of the translational reading frame [37], ultimately leading to the complete absence of the dystrophin protein, which plays a crucial role in proper muscle function [34,36]. A milder form of muscular dystrophy that is also linked to mutations in the dystrophin gene is known as Becker muscular dystrophy (BMD; MIM #300376). Of note, mutations found in BMD patients maintain the translational reading frame and result in a shorter yet partially functional protein [38]. In DMD, the therapeutic intervention aims to restore the dystrophin reading frame, thereby reinstating at least partial expression of dystrophin in DMD-affected muscles and consequently reducing disease severity, similar to what happens in BMD. This is achieved with the design of tailored ASOs, which selectively induce exon exclusion, resulting in the restoration of a correct reading frame [39,40]. Consequently, a shorter yet partially functional protein is produced [40], leading to improved muscle strength and function in affected individuals. In recent years, some attempts to use ES as a therapeutic approach have also been made in cancer. For instance, ASOs have been designed to induce skipping of a specific exon (i.e., exon 4) of the ETS-Related Gene (ERG), which is an oncogene, in prostate cancer cells. This generates an out-of-frame transcript with significant reduction in ERG protein levels, leading in turn to reduced cell proliferation, increased cell death, and reduced cell migration [41]. In a more recent study involving the two isoforms of the PKM (Pyruvate Kinase M1/2) gene, ASOs were used in one case (PKM1 isoform, a tumor suppressor) to restore its expression, and in the other (PKM2 isoform, an oncogene) to decrease it, resulting in reduced tumor growth [42]. In another study, a gene that is often improperly over-expressed in leukemia and solid tumors, the Wilms tumor 1 (WT1) gene, was considered for the ES approach [43]. Specifically, the ASO-mediated skipping of exon 5 resulted in decreased cell viability and survival in leukemia cell cultures [44].

To enhance and broaden these endeavors for a wider range of applications, including undruggable targets in cancer, it is vital to acknowledge the challenges associated with designing ASOs. In particular, numerous design criteria must be taken into account when selecting therapeutic sequences. To assist in this process, some computational tools have been developed to evaluate essential nucleic acid properties and facilitate ASO design. These tools include various functionalities, including estimation of self-complementarity and tendency to form intra-molecular hairpins, as well as the calculation or estimation of molecular weight, solution concentration, melting temperature, and absorbance coefficients [45]. Concerning the study of RNA molecules in particular, software packages such as ViennaRNA Package 2.0 [46] and RNAstructure [47] offer tools for predicting secondary structures and RNA–RNA interactions. Additionally, databases such as SpliceAid 2 [48] provide information on RNA target motifs that are bound by splicing proteins in humans. In particular, a couple of recent web applications and databases have been developed for the design, analysis, and visualization of siRNA and antisense oligonucleotides, PFRED (PFizer RNAi Enumeration and Design tool) [49] and eSkip-Finder [50], offering comprehensive features to aid researchers in their ASO design efforts.

The computational procedure proposed in this study aims to streamline the entire process of designing ad hoc ASOs, from in silico identification of potential target candidates to a list of customized ASOs to induce the intended ES events. Moreover, it takes into account cancer-relevant features, such as mutation frequency in patients, holding special value for the design of ES-based therapeutic intervention in oncology. To this end, our approach incorporates state-of-the-art rules to identify the most promising candidate exons of a gene of interest, which is first classified in silico as an oncogene or a tumor suppressor. Subsequently, specific ASOs are designed following guidelines for morpholinos [51,52], tailored for selected ES events. These events can contribute to desired outcomes, such as generating oncogene variants that lack activity or tumor suppressor variants that could exhibit partially restored functionality. By combining these methodologies, our computational procedure provides researchers with valuable assistance in their efforts to develop innovative therapeutic interventions based on targeted ES.

## 2. Results

### 2.1. Development of an Integrated Computational Procedure to Support the Design of ES-Based Therapeutic Strategies in Cancer

We developed a computational procedure that, given a gene of interest, initially classifies it as either an oncogene or a tumor suppressor, and once all its annotated transcripts are collected, the pipeline proceeds to select potential target exons for ES and return the expected protein variants as the outcome. The pipeline predicts whether the skipping of these exons results in a shortened transcript that remains in frame or shifts out of frame. To further investigate the shortened products, they are translated in silico into amino acid sequences, and those labeled as out-of-frame are subjected to the prediction of their potential degradation through NMD. The procedure then allows for the design and evaluation of candidate ASO sequences to induce the skipping of specific exons that are favorable for achieving the intended goals, such as producing inactive oncogene variants or restoring at least partially functional tumor suppressor variants. The overall procedure is illustrated in Figure 1, and an overview of each individual sub-task is briefly presented in the next subsections.

#### 2.1.1. Classification of a Gene of Interest as an Oncogene or Tumor Suppressor

Based on COSMIC data [10] regarding annotated mutations, the computational procedure enables the in silico classification of a gene as either an oncogene or a tumor suppressor. This classification is based on the principles of the so-called “20/20 rule” proposed by Pavel et al., 2016 [7]. According to this rule, oncogenes tend to have more than 20% of annotated mutations occurring as missense mutations at recurrent positions, while tumor suppressors tend to have over 20% of inactivating mutations (for more details, refer to the Materials and Methods section). Considering this evidence, the procedure automatically assesses the likelihood of a gene belonging to either category, and the classifier assigns the appropriate label, either “oncogene” or “tumor suppressor” (Figure 1A).

#### 2.1.2. Identification of Candidate Exons to Be Targeted for Skipping

Given a gene of interest, the procedure performs a comprehensive screening of all isoforms annotated by GENCODE [53] to identify exons amenable to be skipped (Figure 1B). Only protein-coding transcripts are considered for further investigation. Of all exons in the selected transcripts, those containing the start and stop codons are excluded from the procedure, as well as any non-coding exons that may occur before and/or after them. This exclusion is made to prevent interference with the essential regions responsible for initiating and terminating protein translation [51,52]. To identify candidate exons to induce therapeutic targeted exon skipping, the procedure proceeds to determine the specific outcome by considering the following two scenarios, which may potentially result from ES: (1) in-frame transcript; (2) out-of-frame transcript. Each shortened transcript is then translated into the corresponding amino acid sequence. The skipping of an exon can result in the formation of a PTC, sometimes leading to the degradation of the resulting mRNA through NMD [54]. Thus, the prediction of the occurrence of degradation via NMD is performed for all the out-of-frame transcripts based on the most relevant rules associated to NMD evasion, namely, the so-called “50–55 nt rule”, the “last-exon rule”, and the “start-proximal rule” (see Materials and Methods section for details), which evaluate the position of the PTC formed following ES. The above step in the procedure (Figure 1B) returns the list of all exons deemed potential targets for ES, along with the corresponding shortened transcript and amino acid sequences obtained as a result of their skipping and, for the subset of out-of-frame transcripts, prediction of NMD events.

#### 2.1.3. Analysis of the Mutational Profile

Aimed at the selection of the best candidate exonic targets for the design of ES-based therapeutic strategies, their mutational profile in cancer is next evaluated based on COSMIC data [10]. To this end, the implemented procedure identifies the exons marked by the highest mutational burden, specifically those exhibiting mutations in the largest number of cancer patients. In particular, taking into account only point mutations, the procedure automatically and carefully maps the annotated genomic mutations to the corresponding exon regions [55] and calculates the absolute frequency of mutations within each exon of the gene of interest. This is performed by counting the number of patients who have at least one mutation in a specific exon. Next, the procedure calculates the relative frequency of mutations per exon by computing the ratio between the number of patients with at least one mutation in the given exon and the total number of patients with at least one mutation across the whole gene. Finally, exons are ranked by the decreasing number of absolute mutation frequency, and the top 10 ranking exons are reported to proceed to ASO design (Figure 1C). This step of the computational procedure aims to prioritize ASO design on exons enriched in clinically relevant mutations, while limiting the list of candidate exons to a manageable number.

#### 2.1.4. Design and Evaluation of Ad Hoc ASOs to Induce Desired ES Products

This step of the computational process integrates the sequence of exons that may undergo ES and those that are highly mutated and generates 25-nucleotide-long ASO sequences, which is the recommended length for morpholino oligos [51,52], to induce ES. These ASOs, whose binding is based on the principle of base complementarity, are designed to interact with two types of splicing regulatory sites: splice junction sites (*donor* and/or *acceptor* splice sites) and intra-exon splice sites (Figure 2). For each splice junction, we design a set of 7 ASOs (14 for each exon) to hit the target mRNA at the overlap of the intron–exon or exon–intron border, because this approach has been shown to maximize the efficiency of mRNA splicing alteration using morpholinos [51]. For the ASO sequences designed within exon regions, only those completely overlapping at least one exonic splicing enhancer (ESE) domain are selected for further consideration. The number of ASO sequences drawn within each exon varies depending on the exon’s length and the number of annotated ESE regions therein. Finally, ASO sequences undergo evaluation based on specificity and reference physicochemical parameters, including CG percentage, G percentage, presence of tetra G, and self-complementarity. Only ASO sequences that meet the optimal threshold values for each parameter, as documented in the relevant literature [52], and have a unique match in the genome are selected and included in the output (Figure 1D) (see Materials and Methods section for details). As a point of reference for evaluating the results of our procedure in generating candidate ASOs, we compared them with four available FDA-approved ASO drugs for DMD treatment [29,30,31,32]. Indeed, our procedure successfully identified ASO sequences either identical or closely matched to all drugs, as shown in Appendix B and Table A1.

### 2.2. Application of the Bioinformatic Pipeline to Frequently Mutated Cancer Genes

As a proof of concept to demonstrate the relevance of the entire computational procedure described previously, we applied it to a clinically relevant set of genes. Specifically, we selected the top 10% most frequently mutated genes in cancer patients based on data stored in the COSMIC database (v. 96) [10]. A total of 72 genes were identified, as shown in Figure 3. The following subsections provide insights into the output data and biological knowledge gained by applying the procedural steps outlined in Figure 1 to this cancer-relevant dataset.

#### 2.2.1. Classifying Role in Cancer for Selected Genes

By applying the above-mentioned 20/20 rule [3,7] (see Materials and Methods section for details), the 72 selected genes were classified according to their predicted role in cancer as follows: 25 tumor suppressors, 23 oncogenes, and 24 genes unclassified due to inconclusive scores (Figure 3, inset). To validate our classification, available for 48 genes, the results were compared with two publicly available compendia of oncogenes and tumor suppressors: (1) Futreal et al., 2004 [56], hereafter *MSigDB*; (2) Tokheim et al., 2016 [57], hereafter *Tokheim*. Overall, 35 out of 48 were also evaluated by *MSigDB*, while *Tokheim* assessed 37 of 48, with classification concordance rates of 86% and 97%, respectively. For detailed information, please refer to Appendix A and Figure A1A,B.

#### 2.2.2. Identifying Potential Exon Targets for ES in Selected Cancer Genes

The analysis of the selected cancer genes (N = 72) revealed an average of 10 protein-coding transcripts per gene according to annotations available in the GENCODE database (release 43). Among these transcripts, the majority (9 out of 10) harbored at least one exon predicted to be susceptible to skipping according to the computational procedure (Figure 1B). Overall, these cancer genes had an average number of 26 exons per gene as possible targets. The in silico simulation of ES events demonstrated that the vast majority of these genes (68 out of 72; 94%) could yield both in-frame and out-of-frame transcripts, supporting the potential for designing therapeutic ES strategies for both purposes: inactivation and functional rescue. Furthermore, our computational procedure evaluates possible degradation via NMD for all shortened products generated through in silico ES that are out-of-frame. Specifically, when considering the 23 genes previously classified as oncogenes in our gene compendium, the majority of them (21/23) produced at least one out-of-frame transcript that was expected to undergo NMD.

In summary, the set of cancer genes analyzed exhibited a diverse range of transcript isoforms, many of which contained exons susceptible to skipping. Importantly, a significant portion of these genes had the potential to generate both in-frame and out-of-frame transcripts when subjected to in silico exon-skipping simulations. Additionally, a considerable number of oncogenes in this dataset have the capacity to generate out-of-frame transcripts that are predicted to undergo degradation through NMD.

#### 2.2.3. Analyzing Exon Mutation Profile of the Selected Cancer Gene Set

In the field of precision oncology, it can be valuable to devise therapeutic strategies targeting exons with the highest incidence of recurrent mutations. Therefore, our methodology involves prioritizing exons within the cancer genes selected (Figure 3), according to this criterion. Specifically, we focused on the exons deemed potential targets for ES (from the previous section, 26 exons on average) in each cancer gene. To identify the most mutated exons, we considered the protein-coding exons that had at least one mutation observed across the largest number of cancer patients. By analyzing the mutational profiles using data from the COSMIC database (Figure 1C), we found that these exons showed an average of about 300 point mutations each. Subsequently, the analyzed exons were ranked by frequency of mutations, and the 10 top-scoring exons per gene in this list were selected as the best candidates for ASO design.

#### 2.2.4. Design and Evaluation of the Best Candidate ASO Sequences

As outlined above, the final step in our computational procedure involves the design of ASO sequences that hold promise for inducing desired protein variants (Figure 1D). In particular, to manage computational burden and prioritize experimental feasibility, we focused on designing and providing as output ASO sequences for a selected list of exons for each gene of interest. This list included the top exons that were evaluated as potential targets for exon skipping and were recurrently mutated in cancer patients, with a maximum of 10 exons per gene. A total of 674 protein-coding exons were considered, regarding 72 selected cancer genes (Figure 3, inset), and two sets of customized antisense ASOs were generated (Figure 2). The first set targeted splicing regulatory sites at the splice junctions, and 14 junction-exon ASOs were designed per exon. The second set of ASOs focused on regulatory sites located within the exon itself. Here, an average of approximately 306 intra-exon ASOs were designed per exon. Next, both sets of ASO sequences underwent an evaluation process based on state-of-the-art knowledge regarding optimal values for a series of physicochemical parameters (see Materials and Methods section for details). The ASO design and filtering step yielded an average of 6 ASOs at the splice junctions and 150 ASOs at the ESE sites, representing the best candidate sequences.

### 2.3. Proof-of-Concept Case Studies

#### 2.3.1. Detecting High-Potential Cancer Genes for Effective Targeted ES Intervention

We prioritized genes based on the availability of the exons that were the most suitable for achieving effective targeted interventions. To accomplish this, we assessed the following conditions for each exon in our dataset: (a) Inclusion of the exon among the ten most frequently mutated exons for the corresponding gene. (b) Desired reading frame for the shortened transcript resulting from ES, taking into account the predicted role of the gene in cancer (oncogene or tumor suppressor) and the intended therapeutic objective (inactivation or functional rescuing, respectively). (c) Presence of at least one ASO for each design strategy (targeting splice sites within the exon or targeting splice junctions) that adheres to the recommended physicochemical parameters for effective targeted splicing. Subsequently, we ranked the cancer genes in our compendium based on the decreasing percentage of their exons that met all three criteria mentioned above (Figure 4). In what follows, we provide more detailed information on two exemplary cases, namely, the first oncogene (NRAS) and the first tumor-suppressor (VHL) featured in this ranking.

The NRAS proto-oncogene GTPase (NRAS) has one annotated protein-coding transcript (ENST00000369535.5) consisting of seven exons. From our computational analysis, two exons, namely, exon 3 and exon 4, emerged as potential targets for ES. The therapeutic objective of an ES-based approach would be to induce protein degradation in order to suppress its gain-of-function effect. Skipping either exon would result in out-of-frame transcripts, although our computational procedure predicts that neither of them would undergo degradation via NMD following ES. In terms of mutational profile analysis, both exon 3 and exon 4 rank among the top ten most mutated exons in cancer patients. Particularly, one of these exons (Ensembl ID: ENSE00001751295.1, exon 3) (Figure 5) accumulates mutations in 4052 patients, which accounts for over 60% of the total number of cancer patients with at least one annotated mutation in NRAS. Consistent with its role as an oncogene, the distribution of mutations along the gene sequence is concentrated at specific positions (Figure 5B) [3,7]. Our computational procedure designed a total of 103 ASOs for NRAS exon 3. Among these ASOs, 89 were designed to target regulatory splice sites within the exon, while 14 ASOs were designed to target splice sites at the junctions with flanking introns. After filtering based on physicochemical requirements, the procedure selected a subset of 63 ASOs, which included 52 ASOs targeting internal ESEs and 6 ASOs targeting splice sites at the junctions, with the latter being available only to target the upstream intron–exon junction (Table 1, Figure 5).

The tumor suppressor gene Von Hippel–Lindau (VHL) has a total of six annotated transcripts, with four of them encoding proteins. Using our computational procedure, we identified two specific exons as prime candidates for developing ES-based therapeutic approaches aimed at partially restoring their function by selecting in-frame transcripts. Skipping either of these two exons would result in shortened transcripts that maintain the correct reading frame. Among these exons, one of them (Ensembl stable ID: ENSE00003504189.1, exon 2) (Figure 6) exhibited a higher frequency of mutations in cancer patients, with 447 patients having at least one mutation in this exon, accounting for approximately 24% of the total mutations observed. To target the regulatory splice sites of exon 2, our computational procedure designed a total of 113 ASOs. Among these, 99 ASOs were designed within the exon itself, while 14 ASOs targeted the junctions with adjacent introns. After applying the effectiveness criteria, a subset of 85 ASOs met the requirements, including 47 ASOs targeting internal ESEs and all 14 ASOs designed at the junctions (Table 2, Figure 6).

#### 2.3.2. Harnessing the Potential of Our Pipeline on Well-Studied Cancer Genes: BRAF and TP53

As a second set of case studies, we applied our computational pipeline to analyze one oncogene (BRAF) and one tumor suppressor (TP53), among those most frequently mutated in cancer (Figure 3, inset). For each of the two selected genes, the pipeline identified exons with the potential for being targeted by ES-based therapeutic strategies. The pipeline prioritized these exons based on their frequency of mutation in cancer patients. Table 3 displays the top ten exons that exhibit the highest mutation frequencies for each selected gene. The table also provides the number of candidate ASOs designed for inducing ES at the splice junctions (ASO-J) and within the target exon (ASO-E). Additionally, a flag (IN/OUT) is included to indicate the correctness of the transcript frame following targeted ES, according to the predicted role of the gene in cancer. In addition, Table A2 in Appendix C lists the best candidate ASO-J sequences for both the BRAF and TP53 genes, designed using our procedure to target the splice junctions of the indicated exons.

## 3. Discussion

The development of personalized therapeutic approaches based on molecular tumor profiling has made significant advancements in cancer treatment. However, many cancer driver genes, including frequently altered genes like the RAS family, MYC, and TP53, remain challenging to target with conventional approaches, making them “undruggable”. This poses a major hurdle to oncology drug development. This article aims to present an integrated computational procedure that facilitates the exploration and implementation of ES-based therapeutic strategies for cancer treatment. The developed computational procedure leverages existing knowledge of annotated transcripts and disease-causing mutations for a specific gene of interest. It guides the selection of target exons and the design of ASOs to induce ES. The procedure also provides insights into the consequences of exon exclusion, including the potential degradation of transcripts through NMD. This enables the evaluation of the impact of ES on protein expression and functionality, leading to a deeper understanding of the case of interest. The approach presented is versatile and can support strategies aimed at producing different desired protein variants, such as variants of inactive oncogenes or partially functional restored variants of tumor suppressors.

As a proof of concept, the study focused on the top 10% most mutated genes in cancer and ranked them based on their suitability for ES-targeted interventions. This resulted in a list of the most promising candidate genes for ES-based therapies that contain the highest percentage of exons meeting the following criteria: (a) The exon, when excluded, should result in desired protein variants that are expected to have a therapeutic effect. (b) The ASOs designed for exon skipping should meet recommended physicochemical parameters and have a unique match in the genome, ensuring their effectiveness in inducing exon exclusion. (c) The target exon should rank among the top ten most mutated exons in cancer patients, indicating its clinical relevance. Based on these criteria, NRAS, a member of the RAS protein family, emerged as one of the most promising candidates, with all of its exons being eligible for skipping. Similarly, other genes, such as KRAS, VHL, CALR, GRIN2A, JAK2, FLT3, IDH1, and IDH2, also exhibited a significant percentage of their exons meeting the criteria for optimal skipping targets. In particular, the percentage of optimal exons ranged from over 20% for GRIN2A, JAK2, FLT3, IDH1, and IDH2 to over 60% for KRAS (another member of the RAS family). These findings highlight the potential of these genes as promising candidates for ES-based therapeutic interventions in cancer.

To provide more detailed examples, the study specifically investigated NRAS (an oncogene) and VHL (a tumor suppressor) as notable cases with clinical relevance. Oncogenic NRAS mutations occur in several cancer types, notably melanoma, acute myeloid leukemia, colon and thyroid cancers, and other hematologic malignancies. While attempts have been made to action NRAS for therapeutic purposes by targeting either downstream effectors (e.g., MEK, CDK4/6 [59]) or upstream activators (e.g., STK19 [60]), the NRAS oncogene itself remains currently undruggable and could be theoretically targeted using ES-inducing strategies, as proposed here. Similarly, the inactivation of the tumor suppressor VHL is a major genetic driver of both hereditary and sporadic renal cell carcinoma [61]. Although VHL-defective cancers can be targeted with clinical success by inhibiting its downstream effector HIF1a or VEGF-driven angiogenesis [62], we hypothesize that VHL function could be restored, at least in part, through ASO-mediated transcript modification. The comprehensive in silico approach successfully identified the highest-scoring exon for each gene and designed corresponding ASO sequences to induce its exclusion. Expanding the scope to include the most frequently mutated genes in cancer, the computational pipeline was applied to the extensively studied oncogene BRAF and tumor suppressor TP53. Although small-molecule inhibitors of oncogenic BRAF have proven highly successful for the clinical treatment of several tumor types, atypical (non-V600E) BRAF mutations and BRAF- dependent acquired resistance remain significant challenges and unmet medical needs, which could be targeted using ES-inducing strategies [63]. On the other hand, TP53, one of the most frequently mutated genes in cancer, remains the prototype of an undruggable tumor suppressor, for which no successful therapeutic strategy has been devised [64]. The computational procedure successfully predicted four potential ES events in the BRAF gene. These ES events would result in the generation of out-of-frame transcripts by skipping frequently mutated exons in cancer. The shortened transcripts would likely be targeted for degradation through NMD, thereby reducing the corresponding protein expression level. For each identified exon, the procedure designed an average of six ASO sequences at the exon–intron junctions and nine sequences overlapping ESEs within the exon. Regarding the TP53 gene, two exon candidates for ASO-mediated therapeutic ES events were identified. The aim of these events would be to maintain the transcript frame after exon deletion and potentially restore, at least partially, the biological protein function. An average of four ASOs to induce skipping were designed at the exon–intron junctions, while approximately 53 ASO sequences that cover the ESE regions within the exon were designed internally.

The results obtained by applying the entire procedure to these case studies demonstrate the effectiveness of the strategy in supporting the use of ASOs as innovative therapeutic interventions. This approach aims to induce exon skipping in oncogenes and tumor suppressors, even in cases with significant mutation burdens that are difficult to target with conventional pharmacological methods. It is important to note that the criteria employed in this procedure are independent of tissue and cancer type, adopting an agnostic approach. However, the same computational method can be tailored to specific cases by utilizing mutational profiles obtained from individual patient screenings. This personalized approach has the potential to enhance targeting accuracy and minimize off-target or side effects. While the systematic approach presented in this study allows for wide-ranging applicability across diverse contexts, offering several advantages, there are limitations to consider. In particular, we recognize the value of tailoring this computational method to specific cases, developing further in-depth analysis methods. For example, the current procedure relies on available genomic and transcriptomic data from databases like COSMIC and GENCODE, which may have limitations in terms of coverage and accuracy. In particular, the evaluation of whether a specific splicing-isoform target is generated and subsequently expressed could be more effectively conducted, by its nature, by analyzing expression profile data related to a specific tissue or tumor [65,66,67]. Furthermore, in this context, a more detailed examination of the consequences of the entire spectrum of mutations that can perturb splicing regulatory regions could be considered [68,69,70] (including synonymous mutations, intronic mutations, utilizing ad hoc developed tools [71,72]). Moreover, the potential role of cryptic regulatory sites in influencing splicing outcomes would deserve further study [73]. Finally, it should be noted that the classification of oncogenes and tumor suppressor genes proposed, which in turn determines the type of molecular effect sought through ES, can be debatable from several perspectives: classification methods and available compendia might not cover all potentially relevant genes; a small but non-negligible percentage of genes may be classified differently using various procedures; in some cases, the same gene can function as an oncogene or tumor suppressor, depending on the type of mutation it undergoes (loss vs. gain of function), such as in the case of TP53 [64]. It is important to note that the current approach, prioritizing the most frequently mutated exon in cancer patients, is flexible and can be adapted to different scenarios and therapeutic goals. For instance, in cases where the therapeutic strategy involves inactivating a mutated oncogene, a safer approach to ASO design may entail targeting frame-shifting exons that are possibly not mutated and are located upstream of the most frequently mutated one. This approach would result in the desired outcome of an out-of-frame transcript, reducing the risk that patient-specific mutations might decrease the affinity for the designed ASOs. On the other hand, in more complex scenarios related to the rescuing of function of tumor suppressors, where it is critical to safeguard the healthy allele, the existence of patient-specific mutations may be the key to enable the selectivity needed in the design of allele-specific therapeutic strategies. A notable example that illustrates the relevance of envisioning allele-specific therapeutic strategies is the haploinsufficiency of the tumor suppressor PTEN, frequently mutated in human cancer. Similar to the strategy pursued with DMD drugs, even subtle changes in its expression levels can potentially alter tumor cell behavior [74]. Additionally, the current ASO design approach yields a substantial number of potential candidates per exon, posing challenges to selecting the most suitable molecule sequence for experimental validation. Furthermore, experimental validation is crucial to confirming the actual impact of predicted ASO-induced exon skipping on protein expression and function. For instance, it has been reported that the inhibition of a splice site may trigger the activation of a cryptic splice site, leading to the formation of a transcript with an unexpected architecture [52]. Moreover, the delivery of ASOs to target tissues or cells poses unique challenges that must be overcome as a prerequisite for their effective therapeutic application. Several efforts are currently invested in improving the delivery of ES-based therapeutic drugs through innovative chemical modifications and conjugation with delivery-enhancing agents, such as fatty acids or peptides [19,75,76,77,78]. In conclusion, the integrated computational procedure developed in this study presents a strategy and provides valuable tools for investigating ES-based therapeutic approaches in oncology. ASOs show promise as innovative and personalized therapeutic interventions, particularly for targeting undruggable driver genes. Further experimental validation, optimization, and technological advancements are necessary to fully harness the potential of ASOs as clinically effective therapies. These efforts could pave the way for the effective utilization of ASOs in cancer treatment, making a substantial contribution to the field of personalized medicine in oncology.

## 4. Materials and Methods

The entire computational procedure described in this work was implemented using the Python programming language.

*Cancer genes and mutational annotation data.* To collect the set of test-case genes, we employed mutation data obtained from the publicly available COSMIC database (v. 96) [10]. Specifically, we retrieved the “CosmicMutantExportCensus.tsv” file (from: Data → Downloads → All Mutations in Census Genes). From this extensive dataset, a cohort of 72 genes was selected by identifying the top 10% most frequently mutated genes in cancer patients.

*In silico classification of genes as oncogenes or tumor suppressors.* Mutation annotation data obtained from the COSMIC database were subsequently used to accurately classify in silico the selected cancer genes (N = 72 genes) into the distinct categories of oncogenes or tumor suppressors by applying the well-established “20/20 rule” [3]. Based on this rule, for a gene to be classified as an oncogene, it must exhibit recurrent missense mutations that account for more than 20% of all documented mutations, indicating gain-of-function alterations. On the other hand, to be classified as a tumor suppressor, a minimum of 20% of annotated mutations within the gene should be inactivating. Thus, this rule captures the two main categories of mutations, namely, gain-of-function and loss-of-function mutations, and accounts for their respective frequencies. Specifically, gain-of-function mutations encompass the following types: substitution_missense, deletion_in-frame, insertion_in-frame, complex_deletion_in-frame. Conversely, loss-of-function mutations include substitution_non-sense, deletion_frameshift, insertion_frameshift. To implement the rule in our computational procedure, we followed the methodology outlined by Pavel et al., 2016 [7]. This involves the calculation, for any gene of interest, of two distinct scores based on the mutation annotation data: the oncogene (ONG) and the tumor suppressor gene (TSG) scores. To determine these scores, we started by assessing the total number of variants present in the gene. Subsequently, we computed the frequencies of both gain-of-function and loss-of-function mutations, relative to the total number of annotated variants for the given gene. In particular, the ONG score represents the frequency of recurrent gain-of-function mutations, while the TSG score corresponds to the frequency of loss-of-function mutations. Lastly, based on the criteria summarized in Table 4, the gene was assigned a specific label. Specifically, if the ONG score is greater than 20% and the TSG score is less than or equal to 5%, the gene is classified as an oncogene. Conversely, if the ONG score exceeds 20% and the TSG score exceeds 5%, or if the ONG score is less than 20% and the TSG score is greater than 20%, the gene is labeled as a tumor suppressor. Genes that do not meet either of these criteria are designated as “unclassified”.

To evaluate the accuracy of our classification, we compared our results with two publicly available lists of oncogenes and tumor suppressor genes. The first list was obtained from the Molecular Signatures Database (v. 7.4 MSigDB). This database contains a comprehensive collection of annotated gene sets for use with GSEA (Gene Set Enrichment Analysis) software. The gene set we utilized consists of genes documented in the literature as being mutated and implicated in cancer development (commonly referred to as “cancer genes”). The list was last updated in 2004 [56]. The second list was extracted from the study conducted by Tokheim et al., 2016 [57]. In their work, the classification of genes as oncogenes or tumor suppressors is based on the integration of results obtained with multiple prediction methods.

*ES-target identification.* To identify potential exons suitable for therapeutic ES approaches in the 72 cancer genes under investigation, the following steps were performed for each gene. Firstly, all annotated alternative isoforms were collected using the GENCODE database (release 43; 2 August 2023; GRCh38.p13). Specifically, the GFF3 file “gencode.v43.annotation.gff3” from the Human section (Comprehensive gene annotation, CHR Regions) was retrieved. Additionally, the corresponding sequences of each transcript and their respective exons were downloaded from Ensembl (release 109; 8 February 2023).

*Prediction of degradation via NMD.* We exclusively applied NMD degradation prediction to the shortened transcripts labeled as out-of-frame resulting from the in silico simulation of ES events. This is because out-of-frame transcripts have the potential to generate PTCs, which are relevant for NMD analysis. To determine whether NMD would be triggered for these transcripts, we applied a set of well-established rules associated with NMD evasion. These rules include two canonical ones known as the “50–55 nt rule” and “last-exon rule”, along with a noncanonical rule called the “start-proximal rule” (as illustrated in Figure 7). Under the “50–55 nt rule,” a PTC located less than 50–55 nucleotides upstream of the last exon–exon junction typically does not activate NMD [79,80]. In our analysis, we employed the more stringent threshold of 55 nucleotides, in line with previous research findings [81,82,83,84,85]. The “last-exon rule” of NMD evasion states that PTCs within the last exon are usually not recognized as premature codons because normal termination codons are predominantly found in this exon. Our procedure, therefore, considered PTCs in the last exon not to be subject to NMD [86,87]. Furthermore, we incorporated a notable “noncanonical” rule discovered in cancer data, referred to as the “start-proximal rule” of NMD evasion. According to this rule, the efficiency of NMD decreases within the 5’-most nucleotides of the coding region of a transcript. Specifically, PTCs located approximately 150 nucleotides from the start codon typically do not trigger NMD [79,87,88]. These rules collectively guided our assessment of the potential for NMD activation in the transcripts under analysis.

*ASO design.* The implemented procedure employs two distinct approaches to generate 25-nucleotide-long ASO sequences aimed at inducing ES. The chosen length (25-mer) is indeed the optimal length recommended for the design of morpholino-type ASOs [52]. The first approach involves designing ASOs that target splicing regulatory sites at the splice junctions (*donor* and/or *acceptor* splice sites), while the second approach focuses on splicing enhancer sites located within the exon, also known as ESEs. For each exon, a fixed number of 14 ASOs (i.e., a set of 7 ASOs each for both donor and acceptor splice sites) are designed to target splicing regulatory sites at the splice junctions, while the number of exonic ASOs designed depends on both the exon length and number of annotated ESEs therein. To generate the ASO sequences, a sliding window of 25 nucleotides is used. Specifically, for ASOs designed within exonic regions, the 25-mer window is shifted by one nucleotide at each step, starting from the first position of the exon and ending with the last window within the exon region. Subsequently, only the 25-mer sequences that completely overlap with at least one annotated ESE site are retained for further evaluation of physicochemical parameters. In the case of 25-mer ASOs designed to target the acceptor splice site, a set of 7 ASO sequences that overlap the intron–exon border is collected. The first ASO in this set begins at intronic position −18 (i.e., ASO sequence −18 to +7 from the intron–exon border, where negative and positive numbers indicate intronic and exonic nucleotides, respectively). Starting from there, the next ASO sequences are drawn by sliding a 25-mer window by one nucleotide at the time until reaching the last ASO sequence in the set, which begins at intronic position −12 (i.e., ASO sequence −12 to +13). Similarly, in case of the set of 7 ASOs designed to target the donor site, the first 25-mer window starts at position +12 from the exon–intron border within the exon (i.e., ASO sequence +12 to −13), and this window is shifted by one nucleotide until reaching the beginning of the last ASO in the set, which is exonic position +7 from the exon–intron border (i.e., ASO sequence +7 to −18). Annotated ESE regions were taken from the SpliceAid database [89]. Each binding site in the database is assigned a score ranging from 1 to 10. In our analysis, we focused only on sites annotated with positive scores (i.e., splicing enhancers) and mapped them within target sequences. Concerning ASO nomenclature, we followed the conventions described by Mann et al., 2002 [58].

*Candidate-ASO evaluation and selection.* The designed ASOs undergo evaluation and filtering based on reference physicochemical parameters, including CG percentage, G percentage, presence of tetra G, self-complementarity, temperature of melting (Tm), and specificity. The ViennaRNA Package was utilized to assess self-complementarity, which generates a dot-bracket string notation that indicates both paired and unpaired bases in the predicted ASO secondary structure. Only ASO sequences with a maximum of 16 contiguous base pairs of self-complementarity are selected [46]. Overall, the design of ASOs follows the guidelines provided by Moulton et al. (2008) [52], including optimal values for physicochemical parameters used to select the best candidate sequences. Tm, a crucial parameter that significantly affects the specificity and effectiveness of ASOs, is calculated using three different methods, with reference to the OligoCalc web tool [45]:Basic melting temperature (Tm) (°C),
$$\mathrm{Tm}=64.9+41\left(yG+zC-16.4\right)/\left(wA+xT+yG+zC\right)$$Salt-adjusted melting temperature (Tm) (°C),
$$\mathrm{Tm}=100.5+41\left(yG+zC\right)/\left(wA+xT+yG+zC\right)-\frac{820}{wA+xT+yG+zC}+16.6\log_{10}\left(\left[{\mathrm{Na}}^{+}\right]\right)$$Nearest-neighbor melting temperature (TmNN) (°C),
$$\mathrm{TmNN}=\left(\Delta H-3.4\frac{\mathrm{kcal}}{{}^{\circ}\mathrm{Kmole}}\right)/\Delta S+R\ln\left(\frac{1}{\left[\mathrm{primer}\right]}\right)+16.6\log\left(\left[{\mathrm{Na}}^{+}\right]\right)$$

The thermodynamic parameters were calculated assuming standard conditions (namely, 1M NaCl, pH = 7, and temperature of 27 °C). The nearest-neighbor and thermodynamic calculations were performed as described by Breslauer et al., 1986 [90], but using the values published by Sugimoto et al., 1996 [91]. RNA thermodynamic properties were taken from Xia et al., 1998 [92]. This program assumes that the sequences are not symmetric and contain at least one G or C. The specificity was evaluated by assessing the absence of other potential binding sites in the genome with up to 2 nucleotide mismatches. The Bowtie [93] short-read aligner was used to map ASO sequences against the human genome (release GRCh38) with the following parameters: -v 2 (i.e., map allowing up to 2 mismatches per read alignment); -a (i.e., report all possible alignments). Only ASO sequences that met both the specificity requirement (i.e., unique match in the human genome considering up to 2 mismatches) and physicochemical requirements were reported for any gene of interest as candidates to induce the selected ES products. Recommended thresholds for selected physicochemical parameters, such as CG range (40%-60%), G content (up to 36% G), self-complementarity (16 contiguous H-bonds maximum), consecutive G (maximum of 3 consecutive Gs), and oligo length, were taken from Moulton et al., 2008 [52].

## Figures and Tables

**Figure 1 ijms-24-14862-f001:**
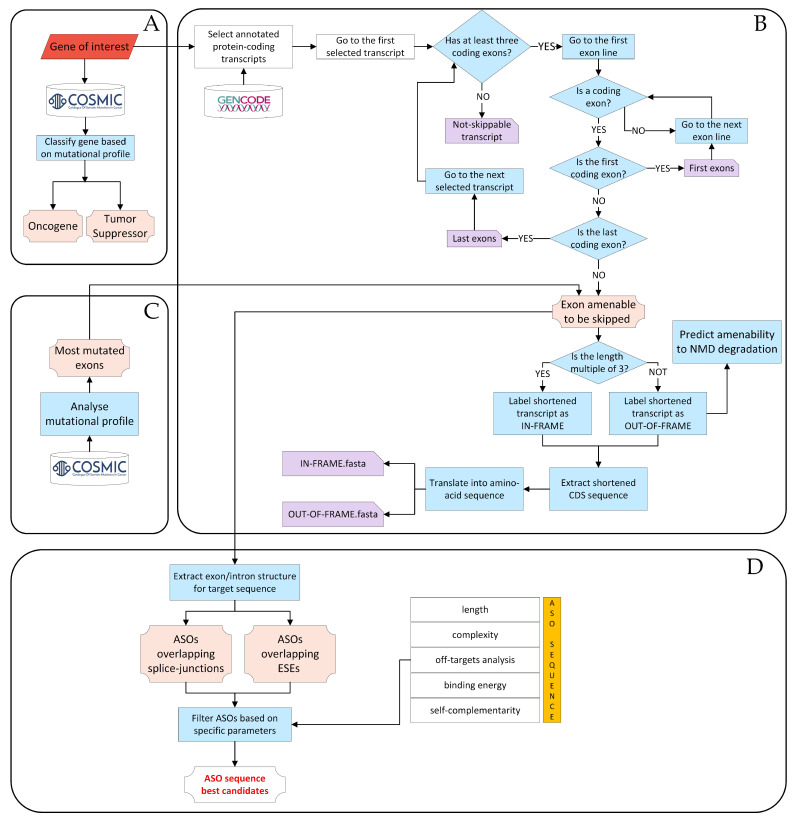
**Computational procedure flowchart.** The procedure begins with a gene of interest and proceeds as follows: (**A**) The gene is classified as either an oncogene or a tumor suppressor by analyzing its mutational profile obtained from the COSMIC database. (**B**) All annotated transcripts of the gene are collected from the GENCODE database. Candidate exons that could undergo skipping are selected, and the corresponding ES events are classified as either in-frame or out-of-frame. For out-of-frame transcripts, the potential degradation through NMD is predicted. (**C**) Among exons that could undergo skipping, the procedure identifies exons that are more frequently mutated in cancer patients based on data from the COSMIC database. (**D**) Candidate ASOs are designed and evaluated. These sequences are differentiated based on whether they overlap splice junctions or bind to ESEs within the exons. Abbreviations: **ASO** = antisense oligonucleotide, **CDS** = coding DNA sequence, **ESE** = exonic splicing enhancers, **NMD** = Nonsense-Mediated Decay.

**Figure 2 ijms-24-14862-f002:**
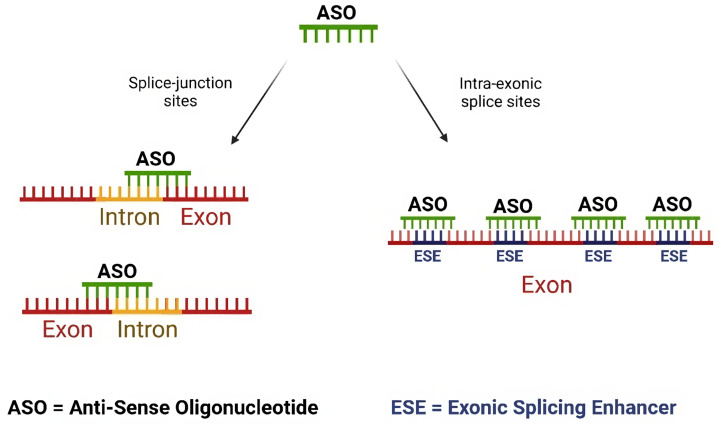
**Schematic representation of the ASO design approaches.** The procedure involves designing ASO sequences for a target exon using two distinct approaches. The first approach (indicated by an arrow pointing to the left in the figure) targets the splice-junction sites, including the donor and/or acceptor splice sites. The second approach (indicated by an arrow pointing to the right in the figure) focuses on the ESE regions within the exon. Figure created with Biorender.com.

**Figure 3 ijms-24-14862-f003:**
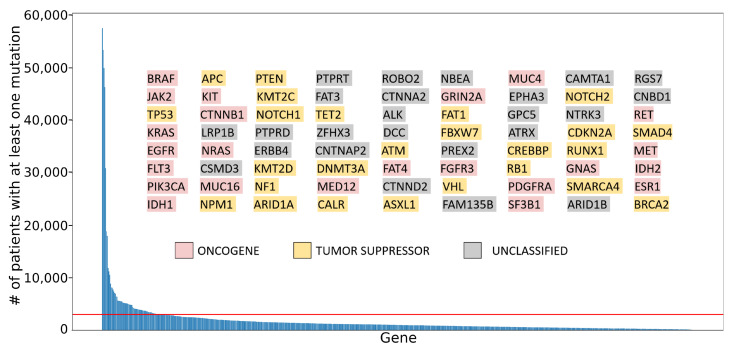
**Most frequently mutated genes in cancer patients.** Genes from the COSMIC cancer gene census are ranked based on the number of cancer patients with at least one mutation (Y-axis) in the indicated gene (X-axis). The horizontal red line indicates the threshold applied for selecting the top 10% of the most frequently mutated genes. The selected genes (inset, N = 72) are highlighted and listed in columns, arranged in descending order of mutation frequency (from top to bottom and from left to right). The color highlighting in the figure corresponds to computationally predicted cancer gene roles, as indicated in the color legend and described in Section 2.2.1 of the main text.

**Figure 4 ijms-24-14862-f004:**
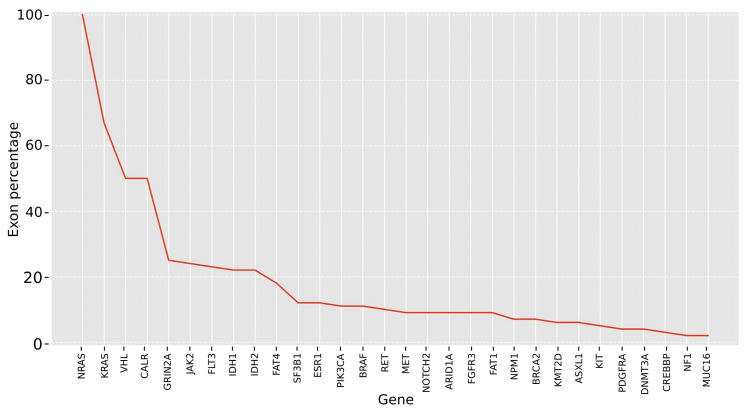
**Ranking of genes based on the percentage of exons that meet all the criteria considered favorable for effective ES-based intervention.** These criteria include the following: (1) Inclusion of the exon among the ten most frequently mutated exons for the corresponding gene. (2) Ensuring the correct frame is obtained following ES, depending on the predicted cancer role (specifically, an out-of-frame outcome for oncogenes and an in-frame outcome for tumor suppressors). (3) Existence of at least one antisense oligonucleotide (ASO) available for both design strategies, which involve targeting splice sites within the exon or at junctions, while adhering to recommended physicochemical values.

**Figure 5 ijms-24-14862-f005:**
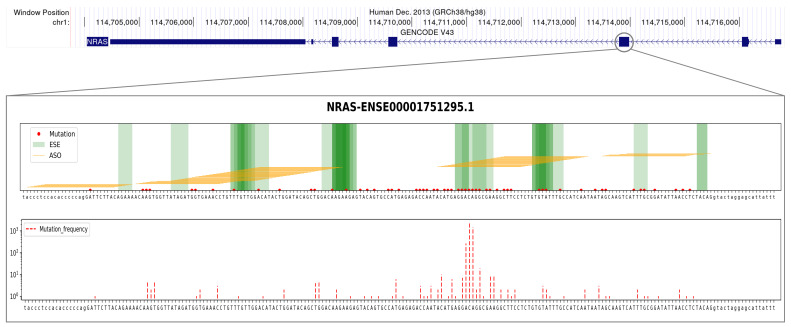
**Example of a suitable candidate for designing an ES-based therapeutic approach targeting the NRAS oncogene.** The figure summarizes pertinent genomic (top panel), sequence (panel **A**), and mutational (panel **B**) information concerning the design of an ES-based therapeutic approach targeting a specific exon (Ensembl ID: ENSE00001751295.1) of the NRAS oncogene that our computational procedure identified as a highly suitable candidate. Details are as follows: The top panel shows the annotated NRAS transcript along with the selected exon highlighted with a grey circle. (**A**) Visualization of 25 nt ASO sequences (represented by orange horizontal lines) designed to target ESE sequences (depicted by green vertical bands) or splice junctions. The objective is to induce the skipping of this particular exon in the mature transcript. The presence of mutations, as annotated in the COSMIC database for cancer patients, is indicated by red dots along the DNA sequence. The uppercase letters represent the exon, while the lowercase letters denote the flanking introns. (**B**) Mutation occurrences within the NRAS exon sequence. This figure panel displays red dotted vertical bars, indicating the number of patients (on the y-axis, using a logarithmic scale) with mutations at the indicated nucleotides. Abbreviations: **ESE** = exonic splicing enhancer, **ASO** = antisense oligonucleotide.

**Figure 6 ijms-24-14862-f006:**
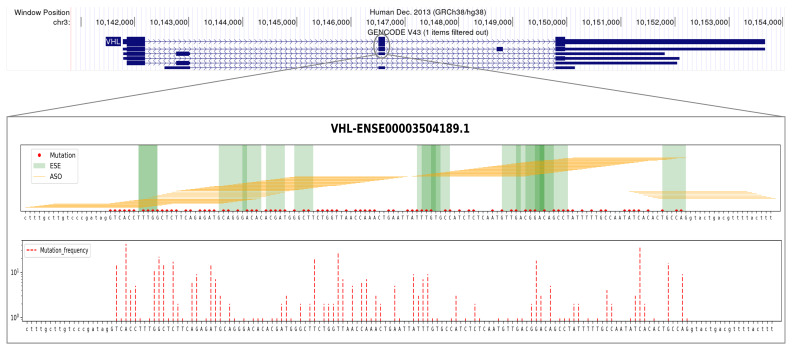
**Example of a suitable candidate for designing an ES-based therapeutic approach targeting the VHL tumor suppressor.** The figure summarizes pertinent genomic (top panel), sequence (panel **A**), and mutational (panel **B**) information concerning the design of an ES-based therapeutic approach targeting a specific exon (Ensembl ID: ENSE00003504189.1) of the VHL tumor suppressor that our computational procedure identified as a highly suitable candidate. Details are as follows: The top panel shows the annotated VHL transcripts along with the selected exon highlighted with a grey circle. (**A**) Visualization of 25 nt ASO sequences (represented by orange horizontal lines) designed to target ESE sequences (depicted by green vertical bands) or splice junctions. The objective is to induce the skipping of this particular exon in the mature transcript. The presence of mutations, as annotated in the COSMIC database for cancer patients, is indicated by red dots along the DNA sequence. The uppercase letters represent the exon, while the lowercase letters denote the flanking introns. (**B**) Mutation frequencies within the VHL exon sequence. The figure panel displays red dotted vertical bars, indicating the number of patients (on the y-axis, using a logarithmic scale) with mutations at the indicated nucleotides. Abbreviations: **ESE** = exonic splicing enhancer, **ASO** = antisense oligonucleotide.

**Figure 7 ijms-24-14862-f007:**
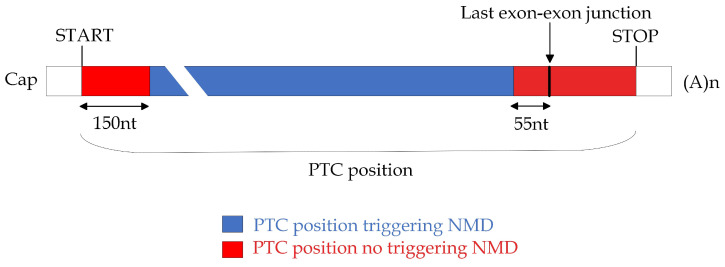
**Rules to predict the occurrence of Nonsense-Mediated Decay (NMD) based on the position of a premature stop codon (PTC).** This figure shows a schematic representation of an mRNA, indicating the positions of translation initiation (START) and termination (STOP) codons that define the main open reading frame. The figure illustrates the application of three rules associated with NMD evasion: the “50–55 nt rule”, the “last-exon rule”, and the “start-proximal rule”. Distinct colors are used to highlight regions where PTCs may be located and the resultant consequences. Specifically, blue shading denotes PTC positions that are more prone to trigger NMD, while red shading indicates positions less likely to elicit NMD. Figure adapted from Carrier et al., 2010 [81].

**Table 1 ijms-24-14862-t001:** **The best candidate ASO sequences designed at the exon 3 splice junctions of the NRAS oncogene.** The table provides the ASOs that were specifically designed at the junctions of the selected exon of NRAS (Ensembl stable ID: ENSE00001751295.1, third exon in the NRAS transcript). The listed ASOs, with their corresponding sequences indicated in the table, successfully passed all physicochemical filters. The nomenclature used for the ASOs follows the conventions described by Mann et al., 2002 [58].

Gene	Exon ID	ASO ID	ASO Sequence
NRAS	ENSE00001751295.1	H3D (13, −12)	UGCUCCUAGUACCUGUAGAGGUUAA
		H3D (12, −13)	AUGCUCCUAGUACCUGUAGAGGUUA
		H3D (11, −14)	AAUGCUCCUAGUACCUGUAGAGGUU
		H3D (10, −15)	UAAUGCUCCUAGUACCUGUAGAGGU
		H3D (9, −16)	AUAAUGCUCCUAGUACCUGUAGAGG
		H3D (8, −17)	AAUAAUGCUCCUAGUACCUGUAGAG

**Table 2 ijms-24-14862-t002:** **The best candidate ASO sequences designed at the exon 2 splice junctions of the VHL tumor suppressor.** The table provides the ASOs that were specifically designed at the junctions of the selected exon of VHL (Ensembl stable ID: ENSE00003504189.1, second exon in the two VHL transcripts). The listed ASOs, with their corresponding sequences indicated in the table, successfully passed all physicochemical filters. The nomenclature used for the ASOs follows the conventions described by Mann et al., 2002 [58].

Gene	Exon ID	ASO ID	ASO Sequence
VHL	ENSE00003504189.1	H2A (−18, 7)	AGGUGACCUAUCGGGACAAGCAAAG
		H2A (−17, 8)	AAGGUGACCUAUCGGGACAAGCAAA
		H2A (−16, 9)	AAAGGUGACCUAUCGGGACAAGCAA
		H2A (−15, 10)	CAAAGGUGACCUAUCGGGACAAGCA
		H2A (−14, 11)	CCAAAGGUGACCUAUCGGGACAAGC
		H2A (−13, 12)	GCCAAAGGUGACCUAUCGGGACAAG
		H2A (−12, 13)	AGCCAAAGGUGACCUAUCGGGACAA
		H2D (13, −12)	AAACGUCAGUACCUGGCAGUGUGAU
		H2D (12, −13)	AAAACGUCAGUACCUGGCAGUGUGA
		H2D (11, −14)	UAAAACGUCAGUACCUGGCAGUGUG
		H2D (10, −15)	GUAAAACGUCAGUACCUGGCAGUGU
		H2D (9, −16)	AGUAAAACGUCAGUACCUGGCAGUG
		H2D (8, −17)	AAGUAAAACGUCAGUACCUGGCAGU
		H2D (7, −18)	AAAGUAAAACGUCAGUACCUGGCAG

**Table 3 ijms-24-14862-t003:** **Top ten mutated exons of the BRAF and TP53 genes.** The table specifically focuses on BRAF as an oncogene and TP53 as a tumor suppressor. For each chosen gene, the table includes the top ten mutated exons and indicates whether the transcript frame following targeted exon skipping is in-frame or out-of-frame (flagged “IN”/”OUT” in the table). The last two columns display the number of candidate ASOs, satisfying all the physicochemical parameters threshold criteria and with a unique match in the genome, designed to induce exon skipping. Columns ASO-J and ASO-E refer to ASOs targeting splice junctions and ESEs within the corresponding exon, respectively. Exons that have at least one ASO sequence available for both the ASO-J and ASO-E design strategies and result in the desired frame in the shortened transcript are shaded in gray to indicate their significance.

Gene	Classification	Top 10 Mutated Exons	IN/OUT Frame	ASO-E	ASO-J
BRAF	Oncogene	ENSE00003485507.1	OUT	4	5
		ENSE00003559218.1	OUT	13	4
		ENSE00003569635.1	OUT	16	0
		ENSE00003587655.1	IN	18	14
		ENSE00001035295.1	IN	34	4
		ENSE00001907699.1	OUT	3	4
		ENSE00003527888.1	IN	12	7
		ENSE00003521664.1	OUT	15	12
		ENSE00003487759.1	IN	13	13
		ENSE00003687908.1	OUT	21	0
TP53	Tumor Suppressor	ENSE00003518480.1	OUT	59	4
		ENSE00003725258.1	OUT	57	13
		ENSE00003712342.1	OUT	30	6
		ENSE00002048269.1	OUT	25	5
		ENSE00003723991.1	OUT	56	10
		ENSE00002073243.1	OUT	25	7
		ENSE00003625790.1	IN	97	5
		ENSE00003670707.1	IN	10	3
		ENSE00003545950.1	OUT	46	5
		ENSE00003786593.1	OUT	18	7

**Table 4 ijms-24-14862-t004:** **Criteria implemented for the in silico classification of genes as oncogenes or tumor suppressors.** The table presents the threshold values, corresponding to the indicated types of mutations, employed to categorize a given gene into different groups (namely, oncogene, tumor suppressor, or unclassified) based on available mutation annotations. These threshold values, which are implemented in our computational procedure, are derived from Pavel et al., 2016 [7].

ONG Score (x/total_mutations × 100)	TSG Score (y/total_mutations × 100)	Classification
>20%	<=5%	Oncogene
>20%	>5%	Tumor suppressor
<20%	>20%	Tumor suppressor
<20%	<20%	Unclassified

x = substitution_missense + deletion_in-frame + insertion_in-frame + complex_deletion_in-frame.y = substitution_non-sense + deletion_frameshift + insertion_frameshift.

## Data Availability

The newly generated data can be found in the manuscript and its Appendix A, Appendix B and Appendix C.

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
