# Peer review of "RNA-Based Strategies for Cancer Therapy: In Silico Design and Evaluation of ASOs for Targeted Exon Skipping"

_ijms, 2023, doi:10.3390/ijms241914862_

Round 1

Reviewer 1 Report

The in silico pipeline proposal by Pacelli and colleagues draws attention to a largely unexplored approach to addressing cancer targets. To date, a relatively small number of studies report antisense strategies to alter RNA structure in cancer; the majority of antisense induce transcript degradation with mostly limited success.

The manuscript is very well written, with a logical structure.

Specific comments

Introduction.

Line 84-96:  While antisense therapeutics for DMD is used only as an example, the summary of DMD presented here is not correct.  DMD is not caused by aberrant exon skipping.  The majority of DMD causing mutations are genomic, frame shifting deletions of one or more exons (>80%), but also splice site mutations, nonsense mutations (10-15%), indels, duplications and other large rearrangements.  The authors are encouraged to more accurately cite the information in reference [29] and to perhaps consider some of the early literature to help their understanding of the aetiology of DMD. Similarly, BMD is most commonly caused by in frame genomic deletions, but also by mutations that impact essential functional domains etc.  Exon skipping does not convert DMD to BMD. Restoring the dystrophin reading frame can induce dystrophin expression in DMD muscle and reduce disease severity.

Results:

Figure 1 and below: Perhaps consider including Splicevault [https://www.nature.com/articles/s41588-022-01293-8] in the analysis; this tool assesses likely consequences of SNPs on splicing [https://github.com/kidsneuro-lab/SpliceVault].  

Line 173: Approximately 25% of all nonsense and missense mutations can alter splicing [doi:10.1101/gr.118638.110].  In one case, multiple different transcripts resulted from a synonymous change [DOI: 10.1136/jmg.2008.059469]. This possibility should be taken into account when considering target selection, since cryptic splicing can deliver gain or loss of function transcript variant, possibly confounding the interpretation of variant function.

Line 176-9: There is some evidence that PTCs in exons closer to the 5’ end of the transcript are less likely to result in NMD than those in exons closer to the 3’ end. Translation of such transcripts to produce a truncated protein possible- but these will likely be degraded. This might be useful to consider when selecting exon  targets.

Line 208: For what reason were ‘up to 18 bases into the intron’ selected?

Figure 2: The figure appears to have been prepared using Biorender.com.  If this is the case, please acknowledge.

Line 262-4, also line 467: Prioritizing oncogene exons with high numbers of reported mutations might exclude otherwise very effective target exons, particularly if they occur upstream of mutation hotspots.  Also, clustered mutations might affect ASO recognition of the annealing site (unless the ASO targets mostly intronic sequence at the splice site) and therefore exclude substantial numbers of potential patients. For this reason, it might be better to target upstream, frame-shifting exons.

Line 487: ‘RNA editing’ is more commonly understood to involve base editing.  Perhaps use the term 'ASO mediated transcript modulation/modification' or similar.

Additional considerations:

The literature shows that donor splice sites are much less frequently found to be effective ASO ES targets, and that mutations near the donor site often result in intron retention. 

Safe and effective nucleic acid delivery to target cells/tissues remains a major impediment to widespread implementation of ASO therapeutics-additional discussion on this matter is warranted.

There is no reference to ASO chemistry- this will impact on ASO design, since phosphorothioate back bone ASO have known toxicities, exacerbated by longer length.  Why was 25 bases selected as the ASO size ?

With effective siRNA drugs entering the market, perhaps a more directed discussion on the potential of the ES strategy to knockdown oncogene expression, and why this might be a preferred approach will strengthen the manuscript.  Also consider the fact that ES in tumour suppressor transcripts will impact the healthy transcript as well as that carrying the mutation.  

Reviewer 2 Report

This study aims to provide a computational pipeline to design antisense oligonucleotides (ASOs) that can inhibit errant exon skipping. The pipeline first predicts potential tumor-related targets and tumor-related skipping sites based on mutational profile, then designs two types of ASO sequences, one overlapping splice-junctions and the other binding to ESEs. Two proof-of-concept case studies, ASOs for NRAS and VHL, are given.

There are two main concerns of this manuscript:

1.     In the in silico design pipeline, the dynamic of RNA folding is not specific considered. RNA folding, including the interaction between ASOs and targets, between ASOs and off-targets region, and self-folding of ASOs, are very important to the efficacy of ASO drugs.

2.     The evaluation is not convincing to show that the design process leads to good ASO drugs. One suggestion is to compare with approved or well-known ASO drugs. For example, Given the same target, would this pipeline be able to generate similar ASO sequences as the approved ones?

Minor issue:

In Fig. 3, the y-axis should be “number of mutations” but not “number of patients”.
